# Early Childhood Diet in Relation to Toddler Nighttime Sleep Duration Trajectories

**DOI:** 10.3390/nu14153059

**Published:** 2022-07-26

**Authors:** Erica C. Jansen, Wentong Zhao, Andrew D. Jones, Teresa A. Marshall, Katherine Neiswanger, John R. Shaffer, Daniel W. McNeil, Mary L. Marazita, Betsy Foxman

**Affiliations:** 1Departments of Nutritional Sciences & Neurology, University of Michigan, Ann Arbor, MI 48109, USA; janerica@umich.edu; 2University of Michigan, Ann Arbor, MI 48109, USA; zwentong@umich.edu; 3Department of Nutritional Sciences, University of Michigan, Ann Arbor, MI 48109, USA; jonesand@umich.edu; 4Department of Community and Preventive Dentistry, College of Dentistry, University of Iowa, Iowa City, IA 52242, USA; teresa-marshall@uiowa.edu; 5Center for Craniofacial and Dental Genetics, Department of Oral and Craniofacial Sciences, School of Dental Medicine, University of Pittsburgh, Pittsburgh, PA 15260, USA; knacct@gmail.com (K.N.); john.r.shaffer@pitt.edu (J.R.S.); marazita@pitt.edu (M.L.M.); 6Department of Human Genetics, Graduate School of Public Health, University of Pittsburgh, Pittsburgh, PA 15260, USA; 7Departments of Psychology and Dental Public Health & Professional Practice, Center for Oral Health Research in Appalachia, West Virginia University, Morgantown, WV 26506, USA; dmcneil@wvu.edu; 8Clinical and Translational Science, School of Medicine, University of Pittsburgh, Pittsburgh, PA 15260, USA; 9Center of Molecular and Clinical Epidemiology of Infectious Diseases, Department of Epidemiology, University of Michigan School of Public Health, Ann Arbor, MI 48109, USA

**Keywords:** sleep duration, diet, early childhood, trajectory

## Abstract

The objective of this study was to evaluate whether dietary habits at age 2 associate with sleep duration trajectories through age 5 in children from north and central Appalachia. A total of 559 children from the Center for Oral Health Research in Appalachia (COHRA) cohort 2 were followed via caregiver phone interviews up to six times between ages 2 and 5. Exposures included data from the year 2 interview: sleep habits, household and demographic characteristics, meal patterns and consumption frequencies of fruits, vegetables, water, juice, milk, and soda. Sleep duration trajectories were identified using group-based trajectory models from ages 2 to 5. Three distinct nightly sleep duration trajectories were identified: *short, increasing duration* (4.5% of the study population); *steady, 9 h of sleep* (37.3%); and *longer, slightly decreasing sleep duration* (58.2%). Using multinomial logistic models that accounted for confounders, children with consistent meal patterns (i.e., meals and snacks at same time every day) and with higher fruit and vegetable consumption were more likely to follow the *longer duration* sleep trajectory compared to the *steady* sleep trajectory. In contrast, children who drank milk more frequently at age 2 were less likely to be in the *longer duration* sleep trajectory than the *steady* sleep trajectory.

## 1. Introduction

Sleep is fundamental for growth and development in young children. Children who do not get adequate sleep for their age are more likely to have lower cognitive development, increased behavioral issues, and higher body weight than those who do get adequate sleep [1,2,3]. At 2 years of age, toddlers are recommended to get 11–14 h of sleep each day (combining nighttime and daytime naps), and by ages 3–5, the recommendation is 10–13 h [4]. Sleep patterns change throughout childhood; therefore, examination of sleep trajectories may be more informative than sleep duration at any one age. Indeed, it is common for children to experience brief periods of problematic sleep [5], but short sleep duration and sleep problems that persist over time are likely more indicative of negative health outcomes [6]. To illustrate, a short sleep duration trajectory from ages 2 through 5/6 within a cohort of French children was associated with hyperactivity/inattention for boys [1]. Among girls, those in the “changing sleep” category had higher hyperactivity/inattention scores. In a cohort of Canadian children, short sleep trajectories from ages 3 months to 2 years were associated with lower cognitive development [2]. Finally, another Canadian cohort study showed that persistent short sleep duration from early to mid-childhood was associated with poor receptive vocabulary during middle childhood [7] and with overweight/obesity [3].

The establishment of bedtime routines has been linked to longer nighttime sleep duration in children [8,9]. Daytime activities and routines are also important for nighttime sleep; for example, lower physical activity and higher screen time during the day have been associated with shorter nighttime sleep [10]. Diet also plays a key role in nighttime sleep. First, meal patterns and timing are known to be important circadian timekeepers (zeitgebers) [11]. Thus, consistent, predictable schedules of eating could be associated with more consistent naptimes and bedtimes and subsequently longer overall duration of sleep. Second, higher diet quality has been consistently linked with better sleep quality and/or longer sleep duration in both adults [12] and children [13,14,15]. Whereas most studies in children are cross-sectional, one study of French children found that a “processed and fast food” diet pattern and a “baby food” diet pattern were associated with a short sleep duration trajectory from age 2 to 5/6 years [16]. 

Disparities in sleep health exist across race/ethnicity groups and in lower- versus higher-income groups in adults and children [17]. The Appalachian region of the US is marked by high rates of poverty in some areas, and higher risk of multiple disease outcomes in both children and adults [18]. Yet, child sleep has not been well-described within this population. Here, we fill this gap using results from a longitudinal investigation of Appalachian children by the Center for Oral Health Research in Appalachia (COHRA), a study originally designed to examine factors influencing oral health. Specifically, we aimed to examine baseline dietary habits in relation to prospective nighttime sleep duration trajectories. 

## 2. Materials and Methods

Study Population: The study sample included 559 children who were involved in the Center for Oral Health Research in Appalachia (COHRA) COHRA2 study [19], a longitudinal investigation designed to collect data on expectant mothers and their babies from pregnancy through the early years of the child’s life. Participants were enrolled in sites in Pennsylvania and West Virginia on a rolling basis between 2012 and 2018 during mothers’ pregnancy, and they and their child were followed at regular intervals. The present study includes child dietary data at 2 years of age and sleep data from ages 2 through 5.5, which was collected as a part of follow-up caregiver phone interviews that occurred approximately every 6 months (a total of eight possible follow-ups in this analysis). Of the 563 children who were enrolled in the study at 2 years of age, 559 of them had dietary and sleep data collected at least once. The study protocol was approved by the institutional review boards (IRB) at the University of Pittsburgh and West Virginia. 

Exposure: During 30-to-45-min interviews conducted by the University Center for Social & Urban Research (UCSUR; http://ucsur.pitt.edu/, accessed on 19 June 2020), caregivers were asked about children’s dietary intake within the last 7 days including typical meal pattern (i.e., whether meals are consumed on a consistent time schedule), fruit consumption, vegetable consumption, water, milk, 100% juice, and soda/pop. Response options for meal patterns were “meals and snacks at about the same time most days”, “meals and snacks at different times most days”, or “snacking throughout the day with few (if any) meals”. For individual foods, response options for frequency of intake were never or once, every few days, once a day, and several times a day. For this study, we used only the dietary information collected at the 2-year-old interview. This decision was made based on the conceptualization of the research question and due to data sparsity in some of the follow-up interviews. 

Outcome: At each phone interview, caregivers were asked about their child’s sleep habits, including nighttime and daytime sleep and presence of sleep problems (yes/no). Daytime sleep duration was defined to be the typical duration of child’s sleep from 7 a.m. to 7 p.m.; nightly sleep duration was the typical duration of child’s sleep from 7 p.m. to 7 a.m. The exact sleep questions were “how much time does your child spend in sleep during the night between 7 in the evening and 7 in the morning?”, “How much time does your child spend in sleep during the day between 7 in the morning and 7 in the evening?”, and “Do you consider your child’s sleep a problem?”. For sleep problems, response options were not a problem at all, a small problem, a very serious problem, and don’t know. Although sleep recommendations for this age group include all sleep within a 24-h period (i.e., nighttime sleep + naps), we decided a priori to focus on nighttime sleep duration, given that circadian-related mechanisms between diet and sleep may operate more strongly at night when melatonin levels are highest.

Confounders: Potential confounders were identified a priori from the literature and included the following variables collected at the 2-year-old follow-up: age in months, ever breastfed (yes/no), whether the child attended daycare, sleep problem (yes/no), mother’s age at birth, gestational week at delivery, child sex, vaginal delivery (yes/no), birth weight (kg), household income, and maternal education. 

### Statistical Analysis

Sleep trajectories were identified using a STATA plug-in for group-based trajectory models (command “traj”). The number and shape of trajectories (linear, quadratic, or cubic) were identified based on model fit statistics (Bayesian information criterion), as outlined previously [20]. Briefly, the trajectory model was built by adding trajectories one at a time, and the linear, quadratic, and cubic functions were tested for each trajectory. To choose the more appropriate model fit at each step, the BIC values for the more complex model were compared to the simpler model using previously established guidelines for weak, moderate, or strong evidence in support of the more complex model. The number of trajectories to retain was also informed by previous literature on child sleep and interpretability (i.e., it is recommended that each trajectory represents approximately 5% of the sample or greater). Each child was assigned to one of the trajectories based on the group trajectory for which they had the highest probability of membership. Children did not need to have sleep data at every follow-up visit in order to be assigned to a trajectory. The majority of participants (84%) had at least three data points for sleep, and exclusion of participants with fewer than three data points did not alter findings.

To examine associations between potential confounders and sleep trajectories, we estimated means ± SD of continuous confounders (proportions for categorical variables) according to sleep trajectory category. To evaluate the primary research question, we used multinomial logistic regression models to compute adjusted odds of nightly sleep duration trajectories according to categories of baseline eating behaviors and adjusted for child’s age, household income, delivery method, mother’s age at birth, and maternal education (variables that were associated with sleep trajectories). 

## 3. Results

At baseline, children were on average (±SD) 2.0 ± 0.1 years of age; 54% were male. The average nighttime sleep duration at baseline was 9.8 ± 1.5 h, and average daytime duration was 2.1 ± 1.0 h. Children were followed four times on average, over a median of 2.3 (1.1, 3.5) years. By the end of the follow-up period, children averaged 5.5 ± 0.1 years of age, had an average nighttime sleep duration of 9.6 ± 1.2 h, and average daytime duration of 0.5 ± 0.9 h. Three distinct nightly sleep duration trajectories were identified (Figure 1): (1) *short, increasing duration (short duration)*; (2) *steady, 9 h of sleep (steady)*, and (3) *longer, slightly decreasing duration (longer duration)*.

The *short duration* group represented 4.5% of the sample and averaged 6.5 ± 1.5 h of sleep at night at baseline, increasing to 7.0 ± 0.7 h of nightly sleep at follow-up. The *steady* group represented over a third of the sample (37.3%) and averaged right around 9 h at night consistently over time. The *longer duration* group represented the majority, starting with 10.6 ± 1.0 h of sleep and declined slightly to 10.2 ± 0.7 h by follow-up. Children in the *longer duration* trajectory (considered as optimal sleep health) were more likely to have been delivered vaginally, to live in higher-income households, and to have mothers with higher education than those in the other groups. In addition, mothers of children in the *short duration* group were much more likely to report that their child had a sleep problem (44% vs. 16% and 15%, respectively) (Table 1). 

Baseline dietary habits were related to sleep duration trajectories (Table 2 and Table 3). After accounting for confounders, children with non-habitual meal patterns (i.e., meals and snacks not taken at same time every day) were more likely to follow the *steady* trajectory compared to the *longer duration* trajectory (optimal sleep health). To illustrate, children who ate meals and snacks at different times each day rather than at the same time each day were two times as likely to be in the *steady* group versus the *longer duration* trajectory (95% CI 1.1, 3.5; Table 3). Similarly, children who drank milk more frequently at 2 years of age were 1.9 (95% CI 1.0 to 3.5) times more likely to be in the *steady* trajectory than the *longer duration* trajectory. In contrast, children with higher fruit and vegetable consumption were more likely to be in the *longer duration* trajectory than the *steady* trajectory. Children who consumed fruits or vegetables several times per day were 0.35 (95% CI 0.188, 0.69) and 0.47 (95% CI 0.27, 0.80) times less likely to be in the *steady* trajectory compared to the *longer duration* trajectory, respectively.

In sensitivity analyses, we evaluated potential confounding by sleep problems and by maternal depression scores (CES-D scale) and found that estimates were not appreciably altered. In addition, we considered the possibility of correlations among the dietary variables. Fruit and vegetable consumption were the only variables with moderate correlation (r = 0.4). The inclusion of both of these variables in one model resulted in an attenuation of the association between vegetable consumption and longer sleep duration trajectory. 

## 4. Discussion

Within this cohort of young children from north and north-central Appalachia, we found that inconsistent mealtimes, higher milk consumption, and lower fruit and vegetable consumption were each associated with a shorter sleep nighttime duration trajectory over time. 

An inconsistent mealtime was related to shorter nighttime sleep. To illustrate, children with meals and snacks at different times every day were over twice as likely to be in a shorter sleep duration trajectory. To our knowledge, this finding has not been reported previously. However, there are several potential mechanisms that might explain this association. First is the content of snacks versus meals. Reports from NHANES have shown that toddlers who consume more snacks consume more calories overall, and these calories are high in carbohydrates and discretionary items with added sugar and sodium [21]. In contrast, toddlers that consume more family meals have higher-quality diets [22]. In turn, higher quality diets have been associated with better sleep among toddlers. Within our dataset, the toddlers who had a snacking pattern also had higher caregiver-reported intakes of chips, desserts, candies, and less fruit. A second explanation could be related to the timing itself. Eating is one regulator of circadian rhythm; thus, keeping a consistent meal schedule would likely reinforce a consistent bedtime and wake time schedule, which is an important aspect of achieving optimal sleep. In adults, inconsistent meal timing is closely aligned with inconsistent sleep timing [23] and to poor cardiometabolic health [24]. One potential mechanism linking meal and sleep times to each other and to metabolic health is through alteration of circadian clock gene expression [25,26]. A third explanation for the association we observed between inconsistent mealtimes and nighttime sleep duration is that there are other behaviors related to snacking behaviors (i.e., confounders), such as higher screen time [27] or lower physical activity, which have each been related to shorter sleep duration in this age group. Household and family-level characteristics also very likely play a role in achieving both consistent meal patterns and consistent nighttime routines. 

Beyond the consistency of meals and snacks, we also found that individual foods were associated with nighttime sleep duration. Specifically, we found that higher fruit and vegetable consumption were associated with longer sleep duration. This finding is consistent with some other studies in toddlers [13,14]. In a French mother–child cohort, a higher score on a “fruits and vegetables” dietary pattern was associated with longer sleep duration among girls [13]. In addition, a study among preschoolers from low-SES households in the US found that higher adherence to a “vegetables, healthy proteins and sides” pattern was associated with less variability of sleep duration from weekends to weekdays and a later timing of sleep (i.e., later bedtimes and/or wake times) [14]. Fruits and vegetables have also been related to better sleep quality among adolescents and adults [28]. One possible explanation is the antioxidant content, which promotes lower inflammation and has been associated with better sleep health [29]. An alternative explanation is that fiber-rich diets high in fruits and vegetables are related to lower consumption of other foods that may hinder sleep, such as saturated fat [30].

The only drink associated with sleep duration was milk, such that more-frequent consumption of milk was associated with shorter sleep duration. This finding is in contrast with a recent report among Mexican adolescents, where milk and 100% juice were each associated with better sleep health (either longer duration or earlier timing), and soda was associated with shorter sleep duration [31]. It is unclear why milk was associated with shorter sleep trajectories in this study, although it is worth pointing out that we did not have information on the quantity of beverages consumed or the time of day that they were consumed. Further, the fat and added sugar content of the milk was not ascertained. Of note, although it was not statistically significantly related, higher consumption of soda was associated with the shorter sleep duration trajectory in the expected direction [32].

Almost one-third of this sample of toddlers (31%) did not achieve the recommended sleep duration for their age, and 16% of caregivers reported that their child had problems with sleep. This is consistent with national averages of children ages 1–5 based on the US National Survey of Children’s Health (33% for children 1–2 years old and 35% for 3–5 year-olds) [33]. The maternal and household correlates of sleep duration trajectories are also worth mentioning. Older mothers and those with vaginal deliveries had children with longer sleep duration trajectories. This is in line with a perinatal study among 619 mother–infant pairs, which found that lower maternal education, prenatal depression, and emergency cesarean birth were associated with shorter infant sleep duration at 3 months [34]. 

This study has a number of strengths, including a longitudinal design with up to eight follow-ups. Further, with an understudied population of children in the Appalachia region of the US, the findings represent a unique contribution to the literature. There are also limitations to consider. The sleep information was parent-reported rather than being an objective assessment. Diet and meal patterns were also parent-reported, and we did not have detailed information on when each type of beverage or food was consumed throughout the day. Furthermore, the diet questionnaire was not a semi-quantitative food frequency questionnaire, which would have provided the typical number of servings consumed for each food and the total energy intake. Thus, we could not investigate whether the amount of each food or drink was related to sleep. Power was also an issue, especially when trying to make comparisons between the shortest sleep duration category with the other sleep trajectories. We also did not have complete dietary data over the follow-up period in order to evaluate changes in diet over time. There were likely some unmeasured confounders, including screen time and physical activity of the children. 

## 5. Conclusions

In summary, we found several baseline dietary correlates of shorter nighttime sleep duration trajectories among toddlers living in households in Appalachia. Namely, inconsistent timing of meals was related to shorter sleep duration over time. Further, higher intake of milk was associated with shorter sleep duration while higher intake of fruits and vegetables related to longer sleep duration. Ultimately, intervention studies that evaluate the impact of meal modifications (both timing and content) in early childhood are needed to evaluate the causal nature of diet and sleep during this developmental period.

## Figures and Tables

**Figure 1 nutrients-14-03059-f001:**
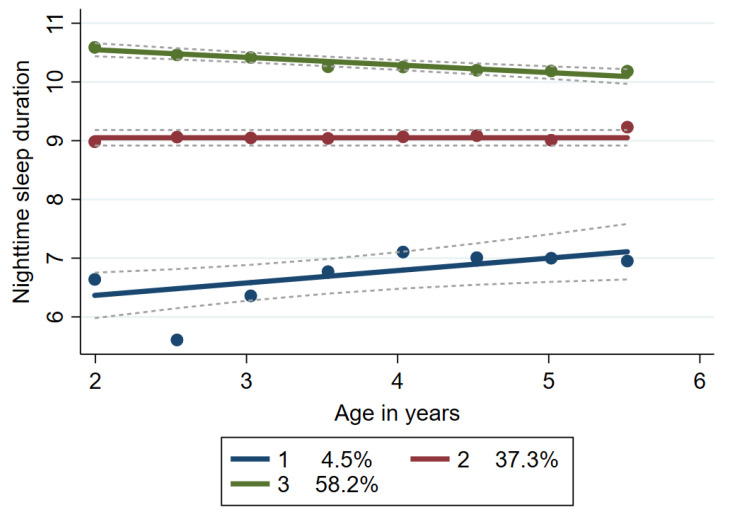
Trajectories of nightly sleep duration from age 2 years to age 5.5 years; 559 children enrolled in Center for Oral Health Research in Appalachia Cohort 2 between 2012 and 2018; 1—short duration; 2—steady; 3—longer duration.

**Table 1 nutrients-14-03059-t001:** Summary of sociodemographic and household characteristics according to trajectories of nightly sleep duration over childhood; 559 children enrolled in Center for Oral Health Research in Appalachia Cohort 2 between 2012 and 2018, with sleep data from age 2 years to age 5.5 years.

	Nightly Sleep Duration Trajectories			
	Group 1 Short Duration n = 25 ^1^	Group 2 Steady n = 201 ^1^	Group 3 Longer Duration n = 333 ^1^	*p* ^3^
Baseline nightly sleep duration	6.5 (1.5) ^2^	8.9 (1.1)	10.6 (1.0)	0.0001
Daytime nap duration	2.9 (1.9)	2.2 (1.2)	2.0 (0.8)	0.10
Age, months	2.00 (0.07)	2.00 (0.06)	1.99 (0.06)	0.79
Breastfeeding, any	0.48	0.39	0.39	0.65
Daycare attendance	0.44	0.44	0.51	0.29
Sleep problem (any at all)	0.44	0.16	0.15	0.001
Mother’s age at birth	28.3 (6.2)	29.7 (5.6)	30.6 (4.4)	0.02
Gestational week at delivery	39.6 (1.1)	39.4 (1.4)	39.5 (1.3)	0.76
Male sex	0.52	0.50	0.57	0.28
Vaginal delivery	0.60	0.66	0.78	0.005
Birth weight, kg	3.38 (0.41)	3.42 (0.50)	3.49 (0.46)	0.19
Household income				<0.0001
$0–14,999	0.24	0.21	0.08	
$15,000–34,999	0.36	0.22	0.11	
$35,000–49,999	0.16	0.11	0.16	
$50,000–74,999	0.16	0.18	0.15	
$75,000–99,999	0.08	0.13	0.23	
$100,000 or more	0	0.14	0.27	
Maternal education				<0.0001
8th grade-completed high school	0.12	0.52	0.36	
Some college/Associate’s	0.07	0.51	0.42	
Bachelor’s degree	0.02	0.26	0.72	
Graduate degree	0	0.23	0.77	

^1^ Largest sample sizes shown. Some individual variables have smaller sample sizes. ^2^ Values represent means (SD) for continuous characteristics and proportions for categorical characteristics. ^3^
*p* values from Kruskal–Wallis tests for continuous variables and chi-squared tests for categorical variables.

**Table 2 nutrients-14-03059-t002:** Eating behaviors at age 2 by to trajectories of nightly sleep duration over childhood; 559 children enrolled in Center for Oral Health Research in Appalachia Cohort 2 between 2012 and 2018, with sleep data followed from age 2 years to age 5.5 years.

		Nightly Sleep Duration Trajectories, *Percentage in Each Category*		
	%	Short Duration	Steady	Longer Duration
Typical meal pattern				
Meals and snacks at same time every day	81.1	3.77	31.93	64.30
Meals and snacks at different times every day	14.0	7.69	52.56	39.74
Snacking throughout the day, few meals	4.9	7.41	51.85	40.74
*p* value		<0.0001		
Fruit consumption				
Never to every few days	9.4	7.69	57.69	34.62
Once per day	19.2	8.41	48.60	42.99
Several times per day	71.4	3.02	29.47	67.51
*p* value		<0.0001		
Vegetable consumption				
Never to every few days	17.8	2.02	51.52	46.46
Once per day	37.8	8.10	33.33	58.57
Several times per day	44.4	2.43	31.58	65.99
*p* value		<0.0001		
Water				
Never or once in last week	3.8	14.29	47.62	38.1
Every few days	3.4	5.26	73.68	21.05
Once a day	14.0	8.97	41.03	50.00
Several times a day	78.9	3.17	32.88	63.95
*p* value		<0.0001		
Milk				
Never or once in last week	14.0	5.13	24.36	70.51
Every few days	5.6	6.45	35.48	58.06
Once a day	14.9	1.20	37.35	61.45
Several times a day	65.7	4.9	38.15	56.95
*p* value		0.239		
Juice				
Never or once in last week	42.8	2.09	31.80	66.11
Every few days	18.6	6.73	34.62	58.65
Once a day	19.5	6.42	31.19	62.39
Several times a day	19.0	5.66	51.89	42.45
*p* value		0.001		
Soda				
Never or once in last week	93.7	4.01	34.73	61.26
At least every few days	6.3	11.43	54.29	34.29
*p* value		0.637		

**Table 3 nutrients-14-03059-t003:** Adjusted odd ratios of nightly children’s sleep duration trajectories according to eating behaviors at age 2; 559 children enrolled in Center for Oral Health Research in Appalachia Cohort 2 between 2012 and 2018, with sleep data followed from age 2 years to age 5.5 years.

	Short Duration vs. Longer Duration, OR (95% CI) ^1^	*p*	Steady vs. Longer Duration, OR (95% CI) ^1^	*p*
Typical meal pattern				
Meals and snacks at same time every day	Reference		Reference	
Meals and snacks at different times every day	1.89 (0.65, 5.49)	0.24	1.98 (1.12, 3.52)	0.02
Snacking throughout the day, few meals	2.23 (0.41, 12.09)	0.35	2.08 (0.83, 5.18)	0.12
Fruit consumption				
Never to every few days	Reference		Reference	
Once per day	0.91 (0.23, 3.63)	0.90	0.65 (0.31, 1.40)	0.27
Several times per day	0.38 (0.10, 1.40)	0.15	0.35 (0.18, 0.69)	0.002
Vegetable consumption				
Never to every few days	Reference		Reference	
Once per day	4.64 (0.97, 22.19)	0.06	0.59 (0.34, 1.03)	0.06
Several times per day	1.15 (0.21, 6.26)	0.87	0.47 (0.27, 0.80)	0.006
Water				
Never or once in last week	Reference		Reference	
Every few days	1.16 (0.08, 17.08)	0.91	3.87 (0.82, 18.19)	0.09
Once a day	0.71 (0.13, 3.78)	0.69	0.79 (0.26, 2.47)	0.69
Several times a day	0.30 (0.06, 1.40)	0.13	0.65 (0.23, 1.86)	0.42
Milk				
Never or once in last week	Reference		Reference	
Every few days	1.66 (0.25, 11.18)	0.60	1.93 (0.73, 5.11)	0.19
Once a day	0.29 (0.03, 2.83)	0.29	1.82 (0.87, 3.82)	0.11
Several times a day	0.99 (0.30, 3.31)	0.99	1.89 (1.03, 3.50)	0.04
Juice				
Never or once in last week	Reference		Reference	
Every few days	2.55 (0.73, 8.88)	0.14	1.04 (0.61, 1.77)	0.90
Once a day	1.58 (0.45, 5.61)	0.48	0.67 (0.38, 1.16)	0.15
Several times a day	1.75 (0.47, 6.53)	0.41	1.58 (0.92, 2.73)	0.10
Soda				
Never or once in last week	Reference		Reference	
At least every few days	2.44 (0.64, 9.37)	0.19	1.89 (0.81, 4.43)	0.14

^1^ From multinomial logistic regression models and adjusted for child’s age, household income, delivery method, maternal age at birth, and maternal education. All factors examined in separate models.

## Data Availability

The data that support the findings of this study are available from the corresponding author upon reasonable request.

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
