# Peer review of "Early Childhood Diet in Relation to Toddler Nighttime Sleep Duration Trajectories"

_nutrients, 2022, doi:10.3390/nu14153059_

Round 1

Reviewer 1 Report

Dear authors,

- Congratulations on your work, which focuses on a trending topic in sleep duration trajectories. The study is well projected and the findings are fascinating. In order to ampliate the introduction section, adding some references to what is present in literature about this topic, I suggest you the reading of the following papers: "Sleep medicine 48 (2018): 194-201", "Behavioral Sleep Medicine 19.3 (2021): 407-425.", and "Sleep Medicine Reviews 49 (2020): 101231.". Since you put the bases for a new paper on the same topic, I suggest you analyze even the patient's psychological aspect regarding this type of trajectory.

- I suggest the authors incorporate some sentences of future perspectives related to the topic in the conclusion section.

- Authors should place the full stop or comma after the reference number in the bracket [ ], not before it. There are also some formatting mistakes in the references section, I suggest the authors check and correct them.

- The English language needs to be improved by minor revisions.

Author Response

- Congratulations on your work, which focuses on a trending topic in sleep duration trajectories. The study is well projected and the findings are fascinating. In order to ampliate the introduction section, adding some references to what is present in literature about this topic, I suggest you the reading of the following papers: "Sleep medicine 48 (2018): 194-201", "Behavioral Sleep Medicine 19.3 (2021): 407-425.", and "Sleep Medicine Reviews 49 (2020): 101231.". Since you put the bases for a new paper on the same topic, I suggest you analyze even the patient's psychological aspect regarding this type of trajectory.

Response: Thank you very much for your review and for bringing our attention to these relevant papers. We have added additional text and references to the introduction as suggested.

For example: “Whereas most studies in children are cross-sectional, one study of French children found that a “processed and fast food” diet pattern and a “baby food” diet pattern were associated with a short sleep duration trajectory from age 2 to 5/6 years[16].”

- I suggest the authors incorporate some sentences of future perspectives related to the topic in the conclusion section.

Response: Thank you for this suggestion. We have added the following:

“Ultimately, intervention studies that evaluate the impact of meal modifications (both timing and content) in early childhood are needed to evaluate the causal nature of diet and sleep during this developmental period.”

- Authors should place the full stop or comma after the reference number in the bracket [ ], not before it. There are also some formatting mistakes in the references section, I suggest the authors check and correct them.

Response: Thank you for alerting us to these issues. They have now been corrected.

- The English language needs to be improved by minor revisions.

Response: We completed a thorough review of grammar and typos, and have made changes.

Reviewer 2 Report

In this study, a total of 559 children was enrolled to evaluate whether dietary habits at age 2 associate with sleep duration trajectories through age 5. Jansen et al. found that several baseline dietary correlates of shorter night-time sleep duration trajectories among toddlers living in households. Inconsistent timing of meals was related to shorter sleep duration over time. Further, higher intake of milk was associated with shorter sleep duration while higher intake of fruits and vegetables related to longer sleep duration. Although the results of this study are important, there are still several issues needed to be addressed.

1.      Since the intake of dietary foods was recorded by frequency, whether the amount of the dietary food shows significant effects on sleeping pattern?

2.      Although the authors indicated that future studies are needed to investigate the mechanisms of these observed relationships, the possible mechanisms including the effects of dietary foods or nutrients on gene expressions related to circadian clock.  

Author Response

In this study, a total of 559 children was enrolled to evaluate whether dietary habits at age 2 associate with sleep duration trajectories through age 5. Jansen et al. found that several baseline dietary correlates of shorter night-time sleep duration trajectories among toddlers living in households. Inconsistent timing of meals was related to shorter sleep duration over time. Further, higher intake of milk was associated with shorter sleep duration while higher intake of fruits and vegetables related to longer sleep duration. Although the results of this study are important, there are still several issues needed to be addressed.

Response: Thank you very much for your review!

  1. Since the intake of dietary foods was recorded by frequency, whether the amount of the dietary food shows significant effects on sleeping pattern?

Response: This is a good question, but unfortunately the diet questions did not ask about the amount or servings of food consumed. We have added this as a limitation.

“Furthermore, the diet questionnaire was not a semi-quantitative food frequency questionnaire, which would have provided the typical number of servings consumed for each food and the total energy intake. Thus, we could not investigate whether amount of each food or drink was related to sleep.”

  1. Although the authors indicated that future studies are needed to investigate the mechanisms of these observed relationships, the possible mechanisms including the effects of dietary foods or nutrients on gene expressions related to circadian clock.

Response: Thank you, this is a great point. We have added some text about the possible mediating role of circadian clock gene expression.

“One potential mechanism linking meal and sleep times to each other and to metabolic health is through alteration of circadian clock gene expression [22, 23].